# The Present and Future of Neoadjuvant Endocrine Therapy for Breast Cancer Treatment

**DOI:** 10.3390/cancers13112538

**Published:** 2021-05-21

**Authors:** Covadonga Martí, José Ignacio Sánchez-Méndez

**Affiliations:** 1Breast Cancer Unit, Hospital Universitario La Paz, 28046 Madrid, Spain; 2Gynaecology Department, Hospital Universitario La Paz, 28046 Madrid, Spain; 3IdiPaz—Instituto de Investigación La Paz, 28046 Madrid, Spain

**Keywords:** breast cancer, endocrine therapy, neoadjuvant, resistance

## Abstract

**Simple Summary:**

The treatment of breast cancer has evolved considerably over the last two decades, leading toward individualized disease management. Hormone-sensitive breast cancers constitute the vast majority of cases and endocrine therapy is the mainstay of their treatment. On the other hand, neoadjuvant or pre-surgical treatments provide a number of advantages for tumor management. In this review we will discuss the existing evidence on neoadjuvant endocrine therapy, as well as its possible future indications.

**Abstract:**

Endocrine therapy (ET) has established itself as an efficacious treatment for estrogen receptor-positive (ER+) breast cancers, with a reduction in recurrence rates and increased survival rates. The pre-surgical approach with chemotherapy (NCT) has become a common form of management for large, locally advanced, or high-risk tumors. However, a good response to NCT is not usually expected in ER+ tumors. Good results with primary ET, mainly in elderly women, have encouraged studies in other stages of life, and nowadays neoadjuvant endocrine treatment (NET) has become a useful approach to many ER+ breast cancers. The aim of this review is to provide an update on the current state of art regarding the present and the future role of NET.

## 1. Introduction

ET is a key pillar in the treatment of ER+ tumors, since it has widely demonstrated its efficacy in improving survival and reducing recurrences. In recent years, the neoadjuvant or presurgical approach has established itself as a very useful strategy in breast cancer management, because it offers numerous advantages. In the first place, tumor downstaging may be achieved, thus increasing breast-conserving surgery (BCS) rates and, in some cases, reducing axillary dissection. On the other hand, assessing the in vivo response enables us to determine drug efficacy, as well as to study any biological or molecular changes that may lead us to explore new biomarkers. Finally, neoadjuvant treatment provides a unique opportunity for validating new treatments, alone or in combination, given that results can be obtained in short periods of time. These advantages have established neoadjuvant chemotherapy (NCT) as a widely accepted approach to estrogen receptor negative (ER−) tumors, but neoadjuvant endocrine therapy (NET) still remains an underutilized tool for ER+ breast cancers, and is frequently relegated to the treatment of elderly or frail patients who are not candidates for chemotherapy.

In this review we will describe the evidence that supports the current use of NET, as well as its potential for future expansion.

## 2. Patient Selection

As is the case with endocrine adjuvant therapy, NET should be considered for luminal-like tumors with ER expression. However, although ET is recommended for tumors even with low ER levels (i.e. <1%) [1,2], NET is normally used only with high ER expression tumors (All red score 7–8), since most trials have achieved better results in these cases [3,4]. Nevertheless, high-ER expression does not always predict a good response to NET. Other factors, such as progesterone receptor (PR) [5] or HER-2 expression, as well as histologic grade, may also influence results [6]. HER-2 overexpression generally discourages the use of NET [5,6], although a few studies (IMPACT, P024) included small subpopulations of HER-2 positive patients and reported over 50% of clinical responses, especially when aromatase inhibitors were employed [7,8].

Regarding histologic subtypes, most studies do not differentiate between ductal and invasive lobular breast cancers in their analysis of response, an exception being a 2011 report confined to lobular breast cancers and published by Dixon et al. [9]. It indicates very good responses in lobular breast cancers alone, with clinical response rates of almost 92% and a mean reduction in tumor volume after three months of 66%.

### Menopausal Status

Most studies concerning NET are confined to postmenopausal women, and this leads most authors and guidelines to reject its use in premenopausal conditions [3,10,11]. Despite this fact, there is evidence of NET response in premenopausal women. The STAGE trial was a single phase III study that enrolled 197 premenopausal ER+/HER-2 negative breast cancer patients who were randomized to either anastrozole or tamoxifen, plus goserelin, for 24 weeks before surgery. The primary endpoint was tumor response assessed by caliper measurement, and this favored the anastrozole arm (70.4% vs. 50.5%), while also more patients in this group benefited from BCS. The authors concluded that anastrozole plus goserelin constitutes a safe choice for premenopausal women with early-stage ER+ breast cancer and can be considered an alternative neoadjuvant treatment [12].

The GEICAM/2006-03 study evaluated differences between NCT and NET in 95 patients with luminal breast cancer, 50% of whom were premenopausal (treated with goserelin plus exemestane). NET was found to be a useful strategy in this group, although responses were higher in the NCT group and this was not observed in postmenopausal women. The authors suggested that these differences might be due to the time it takes to achieve ovarian suppression (four weeks) [13].

Torrisi et al. published a small but very interesting study that included only premenopausal women with locally advanced operable luminal tumors. Patients were treated with letrozole plus triptorelin or goserelin for a median time of four months. Good clinical response (50%, 95% CI 32—68%) and a 50% rate of breast conserving surgery (BCS) were reported. A biological response, in terms of downregulation of the estrogenic signaling, was observed in all patients. The authors considered NET to be a safe alternative in premenopausal women when estradiol suppression is complete [14].

In conclusion, although evidence from clinical trials is scarce, results from other studies show that NET, combined with ovarian suppression, can be considered for premenopausal women, especially when treating luminal A-like tumors (strongly ER+, well differentiated, and with low proliferation rates).

## 3. Clinical Implications of NET

### 3.1. NET vs. NCT

Only three randomized clinical trials have compared NET with NCT. None of them involved a large number of patients, but they all reached similar conclusions.

Semiglazov et al. conducted a phase II study in which 239 postmenopausal patients with stage IIA–IIIB ER+ breast cancer were randomized for three months to either NET (exemestane/anastrozole) or NCT (doxorubicin + paclitaxel). Rates of pathological complete response (pCR) and disease progression did not differ significantly between the arms. BCS rates favored NET and toxicity was higher in the NCT group [15].

The Spanish GEICAM/2006-03 study randomized 95 patients with IHC-defined luminal disease to receive neoadjuvant exemestane for 24 weeks or chemotherapy (four cycles of epirubicin + cyclophosphamide followed by four cycles of docetaxel). Premenopausal women were included in this trial, and they were also treated with goserelin while receiving exemestane. No statistical difference was found between the two arms in terms of clinical response rate, but there was a trend toward a worse outcome in the exemestane arm for premenopausal patients and those with high tumor Ki67 expression [13].

Finally, the NEOCENT trial aimed to compare NCT (epirubicin, 5-fluorouracil, cyclophosphamide) vs. NET with letrozole in postmenopausal women with strongly ER+ breast cancers. Rates of clinical response and radiological response were similar in both arms. Unfortunately, only 44 patients were recruited for this trial, and this weakens the conclusions [16]. The characteristics of these studies are summarized in Table 1.

The meta-analysis published by Spring et al. [17] using data from the abovementioned trials indicated no significant difference in the clinical response rate (OR, 1.08; 95% CI, 0.50–2.35; *p* = 0.85; *n* = 378), radiological response rate (OR, 1.38; 95% CI, 0.92–2.07; *p* = 0.12; *n* = 378), pCR (OR, 1.99; 95% CI, 0.62–6.39; *p*= 0.25; *n* = 378), or BCS rate (OR, 0.65; 95% CI, 0.41–1.03; *p* = 0.07; *n* = 334). However, toxicity was significantly increased in the NCT arm in the three studies.

A recently published meta-analysis including these trials, together with other small studies that compared NCT and NET, found a higher pCR rate with NCT (pooled OR= 0.48; 95% CI, 0.26–0.90), whereas no significant difference in overall response rate (ORR) was observed between those who underwent these two treatment schemes (pooled OR = 1.05; 95% CI, 0.73–1.52) [18].

To sum up, NET does not offer any clear advantages in terms of clinical response or BCS rates when compared to NCT, but it definitely implies less toxicity and better tolerance, thus constituting a safe and effective option for ER+ breast cancers [17,18].

### 3.2. Effect of NET on Breast and Axilla Surgery

#### 3.2.1. Increasing BCS Rates and Enhancing Surgery

NET was first employed in the early 1980s as a therapeutic alternative for elderly women who had inoperable tumors and were unfit for chemotherapy. The primary endpoint of these studies was to evaluate the efficacy of ET as an alternative to surgery, rather than as a neoadjuvant approach. Most of the authors reported response rates over 50%, despite higher locoregional relapse rates. These higher recurrence rates were probably due to the fact that most of these patients did not undergo subsequent surgery, although this did not impact overall survival [19,20,21].

More recent trials involving NET have also demonstrated its value in achieving higher rates of breast conserving surgery and, therefore, an improved cosmesis outcome. In those studies with a treatment duration of more than three months, BCS likelihood can range between 40–80% when using aromatase inhibitors [22,23]. According to Montagna et al. [24], up to 77% of noneligible BCS patients were candidates for breast preservation after NET, although this rate was lower in those with low PR expression.

In parallel with the promotion of BCS, NET has shown itself to be an appropriate alternative for locally advanced, sometimes inoperable, ER+ tumors, [25,26,27] especially with older patients or low grade, luminal A-like tumors. Hence, NET provides a very helpful and safe tool for treating cases with tumors over 2 cm, when there is discordance between tumor and breast size, or in inoperable cases of locally advanced ER+ breast cancers [4].

#### 3.2.2. Axilla Management after NET

It is well established that, after NCT, ER+/HER-2 negative tumors do not achieve a good response in both breast and axilla, with pCR rates below 15% [28,29]. Unfortunately, although axillary-positive patients have been included in most NET trials, very little has been concluded regarding changes brought about by endocrine treatment at this level. Most authors find that NET is less likely to de-escalate surgery in the axilla than in the breast, even though pCR rates range from 1.3% to 11% [24,30]. A slight increase in these rates was observed by Rusz et al., reaching 13% node pCR in N+ patients after one year of NET with letrozole [31]. In a recent review, Stafford et al., after analyzing 4580 ER+ N+ breast cancers from the National Cancer Database, concluded that nodal pCR could be achieved in up to 14.5% of patients [32]. As reported by other authors, lobular breast cancer had a significantly lower percentage of nodal pCR when compared to ductal histologies (12.4% vs. 15.26%, *p* < 0.05) [32,33]. Longer treatment duration (>6 months) seems to increase the likelihood of nodal pCR [24,31].

Axillary management after NCT has been a controversial issue in the past few years, mainly because of the concern that disease may be left behind if an axillary dissection (AD) is not performed. Three trials have proven sentinel node biopsy (SLNB) to be a safe technique when cN1 axilla changed to cN0 after NCT [34,35,36], and this has brought about a widespread change in practice patterns, with a general avoidance of AD when a complete response is presumed. Nevertheless, data regarding surgical management of the axilla after NET is limited. As previously stated, good responses in the axilla after NET are rarely expected, so this might create more difficulties when carrying out a SLNB. Paradoxically, most surgeons seem to apply the same criteria to NET as they do to NCT [37]; however, the context of NCT and, above all, of the high-risk tumors treated in this way, does not seem applicable to NET, since any low-burden disease left in the axilla after NET may not significantly impact the prognosis, as pointed out by Kantor et al. [37,38] These authors concluded that axillary management strategies employed with upfront-surgery patients may also be used with NET patients, which would mean that Z0011 criteria for avoiding AD may be safely applied.

In conclusion, axillary response is rather uncommon after NET, although the prognostic significance of axillary burden after treatment is not different from that of patients managed with upfront surgery.

#### 3.2.3. Surgery Avoidance in Frail Patients

As mentioned above, the initial studies of NET were developed among elderly and/or frail women cohorts. Tamoxifen then arose as a possible alternative to surgery, with clinical responses in over 50% of the patients. As most of the cases did not complete the treatment with surgery, a large number of recurrent events was reported, although this did not affect survival [17,18,19]. In 2007, the Cochrane Library published a review comparing surgery, with or without tamoxifen, with tamoxifen alone for older women with operable breast cancer. When surgery with adjuvant endocrine therapy was compared to endocrine therapy alone, there was no significant difference in overall survival (HR: 0.86), but there was a significant difference in progression-free survival (HR 0.65) for surgery plus endocrine therapy vs. endocrine therapy alone [39]. Later studies using aromatase inhibitors have yielded similar results [40,41], with no survival benefit in spite of an increase in risk progression. In the review performed by Morgan et al. [42], randomized and nonrandomized controlled trials were included that recruited women over 70 years of age with operable breast cancer and treated with NET. A long median time to progression (~49 months), and a quite long duration of clinical benefit (~30 months) were reported, which, together with its low toxicity, make primary endocrine therapy a useful treatment strategy in selected patients. The authors concluded that NET can be an option in cases of short life expectancy (<2 years) and, given their greater efficacy, aromatase inhibitors should be preferred over tamoxifen [42]. Tumor cryoablation or surgery under local anesthesia can be considered in some cases when a tumor reduction accomplished with NET makes a less extensive resection possible [43,44,45].

In other words, although surgery avoidance should not be considered the standard of care, it can be contemplated in the case of a specific group of elderly or frail patients with ER+ breast cancer who offer considerable possibilities of response or disease stabilization with NET.

## 4. Choosing the Best Endocrine Agent

### 4.1. Tamoxifen versus Aromatase Inhibitors

The third-generation aromatase inhibitors (AIs), anastrozole, letrozole, and exemestane, are nowadays considered the standard adjuvant treatment for women with ER+ breast cancer because of their advantages over tamoxifen, and so studies have been conducted in order to confirm their superiority in the neoadjuvant setting.

Four phase III randomized trials have studied the differences between tamoxifen and AIs in that setting (Table 2). Three of these studies were conducted in postmenopausal women and one in premenopausal women.

P024 was a four-month, double-blind randomized trial that compared letrozole 2.5 mg with tamoxifen 20 mg in the preoperative treatment of 337 postmenopausal women who had ER+ locally advanced breast cancer and were ineligible for BCS. Letrozole showed a higher objective response (55% vs. 36%, *p* = 0.0001), as well as a better BCS rate (45% vs. 35%, *p* = 0.02) [7].

Two trials compared anastrozole with tamoxifen. The IMPACT study randomized 330 postmenopausal women with operable or potentially operable ER+ breast cancer to receive either tamoxifen 20 mg, anastrozole 1 mg, or a combination of both [8]. No differences in objective response were found among the three arms, although BCS rates were double in the anastrozole arm in patients who were deemed eligible for BCS by their surgeons (46% vs. 22%, *p* = 0.03). No benefit was obtained from the combination of letrozole and tamoxifen. The three-month PROACT trial compared anastrozole vs. tamoxifen in 451 postmenopausal women with ER+ large, operable (T2-3/N0-2) tumors, with or without chemotherapy [46]. Improvement in feasible surgery at baseline to actual surgery at three months was found to be numerically higher in the anastrozole group compared with the tamoxifen group, although this difference did not reach significance. However, significant differences were observed favoring the anastrozole arm in BCS rates of patients who did not receive chemotherapy (43% vs. 30.8% *p* = 0.04).

Finally, exemestane efficacy was also reported by Semiglazov in a randomized trial with 151 postmenopausal women [47]. Tamoxifen or exemestane were given preoperatively for three months. Significant differences were found in terms of objective response (exemestane 76% vs. tamoxifen 40%, *p* = 0.05) and in BCS (37% vs. 20%, *p* = 0.05), with a shorter response observed in the exemestane arm (57 days vs. 70 days, *p* < 0.05).

A meta-analysis of these trials demonstrated that preoperative use of AIs was more effective than tamoxifen. Pooled results for clinical efficacy revealed a clinical response rate (OR, 1.69; 95% CI, 1.36–2.10; *p* < 0.001), radiological response rate (OR, 1.49; 95% CI, 1.18–1.89; *p* < 0.001), and BCS rate (OR, 1.62; 95% CI, 1.24–2.12; *p* < 0.001) favoring Ais [17].

The only study comparing AIs with tamoxifen in the premenopausal condition was the aforementioned STAGE trial. This showed anastrozole to be superior to tamoxifen when both were combined with goserelin in premenopausal women with ER+ breast cancer [12]. Therefore, AIs constitute a first-choice option in neoadjuvant therapy, just as they already do in adjuvant treatment.

### 4.2. Comparison of Aromatase Inhibitors

The Z1031 trial is the only one to have compared the three third-generation AIs. This was a phase II study that randomized 377 postmenopausal women with clinical stage II/III ER+ breast cancer to receive an AI (anastrozole, letrozole, or exemestane) for four months before surgery [48]. The authors concluded that neoadjuvant AI treatment markedly improved surgical outcomes with no significant differences in clinical, biological, or surgical outcomes among the three agents. It is worth highlighting that half of the patients that were candidates for mastectomy benefited from BCS after NET. Consequently, the three commercially available AIs are all equally effective in neoadjuvant therapy, as is the case when they are used in adjuvant and metastatic treatment.

### 4.3. Selective Estrogen Down -Regulators versus Aromatase Inhibitors

Fulvestrant has been compared to anastrozole in two trials. Quenel-Tueux et al. randomized 108 postmenopausal women who had ER+ breast cancer and were not initially eligible for BCS to receive 500 mg fulvestrant or 1 mg anastrozole for six months [50]. No significant differences were found between the drugs, with similar objective responses (58.9% anastrozole vs. 53.8% fulvestrant) and BCS rates (58.9% in the anastrozole arm vs. 50.0% for fulvestrant).

With very similar results, the phase II UNICANCER CARMINA trial compared fulvestrant with anastrozole in 116 postmenopausal women with ER+/Her-2 negative, operable breast cancer [49]. Clinical response rates at six months were 52.6% (95% CI, 41–64%) for anastrozole and 36.8 % (95% CI, 25–49%) for fulvestrant. BCS was performed in 57.6% in the anastrozole arm and 50% in the fulvestrant arm. Relapse-free survival rates at three years were comparable (94.9% for anastrozole and 91.2% for fulvestrant). Both agents demonstrated effectiveness and were well tolerated, although results were slightly better with anastrozole.

## 5. Treatment Duration

NET duration is usually established at 3–6 months, based on the regular duration of NCT, although the optimal period for this therapy is not yet clearly determined. Several trials have been conducted with the aim of defining an appropriate duration (Table 3). Most of these studies found that a prolongation of NET produced extra benefits [31,51,52,53,54,55,56].

The first study was published by a German group led by Krainick-Strobel et al., concerning a cohort of 32 postmenopausal patients who received letrozole for a period of four months (early responders) to eight months (slow responders) [51]. At diagnosis, all patients were unsuitable for BCS, but the majority achieved sufficient response for BCS after four months, although the objective response rate reached 72.4% after eight months. Similarly, Dixon et al. compared letrozole for periods of three or more than three months [52]. After three months almost 70% of the patients had responded and 60% of them received BCS. With a longer treatment (>3 months), tumor downstaging reached 83.5%, and BCS rates 72%. A Spanish phase II trial recruited 70 postmenopausal women treated with letrozole for 3–12 months [53]. 6.8% of the patients achieved an objective response, 51.8% being partial and 25.0% complete. The median time to objective response was 3.9 months (95% CI, 3.3–4.5) and the median time to maximum response was 4.2 months (95% CI, 4.0–4.5), although 37.1% patients achieved the maximal response within 6–12 months. Despite these results, BCS rates reached only 43%. Another study, one that continued NET up to 12 months, stated that the optimal duration for obtaining BCS was a median of 7.5 months [54]. One Italian trial randomized 120 postmenopausal ER+ breast cancer patients to receive letrozole for 4, 8 or 12 months [56]. Clinical response, as well as pCR and BCS rates, were analyzed. After four months, only 2.5% pCR was observed, although BCS was feasible in 80% of the patients. After 12 months, pCR rates reached 17.5% and almost 90% of the women benefited from BCS. Slightly lower pCR rates (14.3%) are published by Rusz et al. [31], after a one-year treatment with letrozole, although they report relatively modest BCS rates (45%). Consequently, low pCR rates attributed to NET may be due to treatments that are not sufficiently long.

Only two trials have assessed the ideal duration of NET employing exemestane. The study conducted by Fontein et al. (TEAM IIA) [55] reported a significant increase in objective response rate with longer treatments (58.7% <3 months vs. 68.3% >3 months, *p* = 0.031), as well as an improvement in the feasibility of BCS (61.8% vs. 70.6%, *p* = 0.012). On the other hand, the Japanese PTEX46 trial [57] did not find significant differences between treating with exemestane for four months or six months, either in objective response rates (42.3% four months vs. 48% six months, *p* = 0.89) or in BCS rates (50% four months vs. 48% six months).

Because of this variability, clinical guidelines and expert consensus do not specify a recommended duration for NET. It was only at the 2013 St. Gallen Conference that the majority of the panelists supported the use of NET until maximal response [58].

Overall, even though the findings of the studies are mixed, it can be concluded that good responses and BCS rates can be achieved with a NET duration of six months or less, but higher pCR rates and objective responses may be obtained with up to one year of treatment.

A major concern raised when NET is maintained for long periods is the possibility of disease progression due to endocrine resistance. Studies of elderly and/or frail women with ER+ breast cancers who had been treated exclusively with tamoxifen found no differences in overall survival when compared with women treated with surgery, although regional control was poorer in the nonsurgical cohort [59,60].

Interestingly, several of these trials found no difference at all between the surgical and nonsurgical cohorts for the first 18–24 months. Willsher et al. also observed that a partial or complete response at six months was a strong predictor of excellent control at three years [61], especially with highly ER+ tumors. Hence, with an exclusively NET approach, resistance is unlikely to appear during the first year. Beyond 12 months, NET can be maintained in selected cases of elderly women [26], in particular those with a documented early tumor response.

## 6. Monitoring Response to NET

For their primary analyses, adjuvant trials typically employ survival goals (disease-free or breast cancer mortality). However, in the neoadjuvant setting the response criteria are based on the dynamic changes produced during treatment. These may be changes in surgical outcome, tumor size, or histopathological characteristics after therapy.

### 6.1. Assessing Response by Imaging

As with NCT, clinical examination and radiological control are of great importance. The combination of both clinical exploration and imaging constitutes the usual method for assessing response. Unfortunately, the precision of clinical breast examination for determining pCR in patients with locally advanced breast cancer after NET or NCT is only 57%, which is inferior to mammography (74%) and ultrasound (79%) [62], mainly because of the clinical changes (post-therapy fibrosis, undefined tumor margins) produced by treatment. A combination of mammography plus ultrasound (US) appears to be a very reliable way of checking response (80% likelihood) [63], although little has been published concerning these two techniques and NET evaluation.

Based on experience with NCT, many authors recommend MRI as the best procedure for assessing NET response, since it is a dynamic technique [63,64,65,66] that enables changes within the tumor to be verified. Estimating final pathology size after treatment is crucial when employing neoadjuvant therapies, since it allows a surgical approach to be planned. Correlation between post-treatment size and final pathology size is better with MRI than with clinical exploration [67], but no studies comparing MRI accuracy with US or US-mammography are available in the NET setting, although in the NCT scenario some authors have found US to be as accurate as MRI [68]. On the other hand, MRI could provide not only data on responsiveness to therapy but could also predict which patients would benefit most from it. In this sense, minimal or mild background parenchymal enhancement was described as a predictive factor of good response to NET in a small sample published by Hilal et al. [64]. Changes in contralateral breast parenchymal enhancement on MRI during NET might also differentiate between patients with a good prognosis and those with an intermediate/poor prognosis at final pathology [69].

The introduction of novel quantitative functional imaging, such as advanced MR or radionuclide imaging, may be also useful for evaluating changes caused by therapy response [63]. Metabolic changes produced by NET can be assessed using FDG-PET/CT scans. These have been proven to be a helpful tool since there is a correlation between tumor SUVmax and cell-cycle response (Ki67 levels) [70,71] and they may even be considered as a surrogate marker for monitoring tumor response [71]. Nevertheless, there are some limitations to this technique, as low-proliferation tumors (luminal A-like, lobular), which otherwise are the best candidates for NET, have a lower glucose metabolism [72] and this may affect the interpretation of the response. An interesting alternative would be to employ ^18^F-fluoroestradiol (^18^F-FES) as radionuclide [73,74,75]. ^18^F-FES PET/CT can measure the in vivo binding of estrogens and thus can be used to assess the biological activity of ER. Earlier trials reported a high positive and negative percentage agreement of ^18^F-FES PET (87% and 91%, respectively), with ER determined by immunohistochemical testing [76]. Other studies have indicated that FES avidity can be a pharmacodynamic biomarker for ER-directed therapy and could therefore predict a better response to NET and no benefit from NCT [74]. Unfortunately, these novel radiotracers are still not available in most centers and remain at the research stage, although they look very promising as an option in the NET setting. The combination of techniques such as MR imaging and FDG PET-TC, which are available in only a few facilities, might offer advantages for primary breast cancer therapy response assessment [63,73], yet further investigation is required in this field.

Therefore, monitoring NET response may be performed reasonably well with traditional procedures like US or US-mammography, although dynamic or functional techniques can provide greater accuracy in this evaluation.

### 6.2. Pathological Complete Response

Pathological complete response is a validated measure of long-term outcomes in NCT studies, mainly in biologically aggressive phenotypes such as triple negative or HER-2 positive breast cancers [77]. Nevertheless, pCR rates in ER+ breast cancers are very low with both NCT and NET [17], although this does not impact survival as much as happens with non-luminal tumors [77]. Consequently, pCR is not an optimal indicator of outcomes and other biomarkers must be used when validating NET.

### 6.3. Ki67

Ki67 is a nuclear antigen expressed in proliferating tissues, in all cell-cycle phases. It is absent in quiescent cells, and correlates well with other markers of proliferation such as the S-phase fraction or mitotic index, [78] thus being considered a surrogate marker of cell proliferation. Neoadjuvant treatments can induce a reduction in Ki67 value, and such changes have been identified only a few days after the initiation of NET [8,79], indicating a cell-cycle arrest induced by therapy [10]. But in addition, the changes that appear in Ki67 with NET constitute a biomarker with prognostic and predictive power [8,12,44,46], much more precisely related to long-term results than baseline Ki67 [80]. A Ki67 > 10% after two or four weeks of endocrine therapy has been suggested as a cutoff for early identification of non-responders with increased risk of relapse [81], and it is actually considered by many authors in various trials as the necessary threshold for maintaining or withdrawing NET [79,81,82,83,84,85]. Early Ki67 changes produced by NET served as the basis for the trial of PeriOperative Endocrine Therapy: Individualizing Care (POETIC) [79]. This “window of opportunity” study, with over 4300 women recruited, randomized patients to receive either an AI for two weeks before and two weeks after surgery, or no treatment at all. Although this short-course treatment did not improve outcomes, it clearly identified three groups of patients: those with low Ki67 levels before and two weeks after treatment, with a five-year recurrence risk (RR) of 4.3% (95% CI, 2.9–6.3); those whose Ki67 was initially high but became low after two weeks, with a RR of 8.4% (95% CI, 2.9–6.3); and finally, those with permanently high Ki67 scores (RR: 21.5% 95% CI, 17.1–27.0). The authors concluded that patients with low Ki67 or low post-AI-induced Ki67 probably do well enough with standard endocrine therapy, whereas those with high Ki67 scores might benefit from further adjuvant treatments. Interestingly, tumors that do not lower Ki67 after 2–4 weeks also fail to respond well to chemotherapy, indicating an intrinsic endocrine resistance, which makes it essential to consider alternative therapies [81]. The PerELISA trial also found that a Ki67 reduction could also serve to identify subgroups of ER+/HER-2 + patients who had a better prognosis, and for whom ET plus anti-HER-2 treatments, without chemotherapy, could be sufficient [86].

Ki67 presents certain weaknesses as a biomarker, such as intratumor heterogeneous expression, perhaps underrepresented in the biopsy, as well as interobserver variability when quantifying Ki67 staining [87]. Regardless of this, the clinical validity of Ki67 is fairly well established, and it is a useful and accessible tool that may make it possible to distinguish between endocrine sensitive or resistant tumors [87] and, in addition, determine a prognosis.

### 6.4. PEPI Score

Data from the follow-up of the P024 trial enabled Ellis et al. to develop a prognostic model that incorporated standard pathologic staging variables and “on-treatment” biomarker values. In the multivariable analysis, the four factors that showed a significant impact on survival (either relapse-free survival or breast cancer-specific survival) were: pathological tumor size, pathological node status, ER status, and Ki67 natural log intervals—all derived from the surgical specimen analysis. They were later reanalyzed to derive a final hazard ratio associated with each factor. This scoring process generated three risk groups (risk score 0, 1–3, and ≥4) that were associated with relapse risks of 10%, 23%, and 48% (*p* < 0.001) and breast cancer deaths of 2%, 11%, and 17% (*p* < 0.001). This was named the Preoperative Endocrine Prognostic Index (PEPI score (Table 4) [88]. An independent validation of this model was later performed with data from the IMPACT trial, with more favorable outcomes being shown in patients with T1 tumors and a PEPI score of 0; i.e., with a Ki67 score of less than 2.7% and ER expression. After two weeks, the PEPI score may be considered a more robust marker of response than Ki67 in some tumors, especially those with low pretreatment Ki67 or with a Ki67 rebound after the first 2–4 weeks [87]. With specific regard to its clinical application, the PEPI approach could be analogous to pCR with an escalation of therapy for those patients with PEPI > 0 and a de-escalation (avoiding chemotherapy) for patients with PEPI-0 status [82]. A modified PEPI score, including primary PR expression (PEPI-P), was described by Kurozumi et al., with better correlation with breast cancer-specific survival and relapse-free survival than PEPI alone [89]. One inconvenience of the PEPI score is that nodal involvement is a qualitative measurement, so a positive axilla always implies a PEPI score ≥ 3, regardless of whether one or more than three nodes are involved, and so an equivalence between PEPI ≥ 3 and chemotherapy should not always be categorically assumed.

### 6.5. Other Biomarkers

Although Ki67 and the PEPI score are useful and available in clinics, more accurate biomarkers are being investigated. In a small prospective French study, high levels of the *ZNF217* gene were found to be significantly related to reduced responses to NET and independent of Ki67 expression. [90] In another study, a higher expression of the nonproliferative genes *KRAS, CUL2, FAM13A, ADCK2*, and *LILRA2* was significantly associated with tumor shrinkage, and *KRAS, MMS19*, and *IVD* were related to a lower PEPI score (≤3) and, again, this gene expression had no correlation with Ki67 reduction [91]. This seems to indicate that gene expression may eventually provide a more accurate biomarker; however, more research is still needed. 

## 7. NET and Targeted Therapies

Good results with targeted therapies combined with ET in locally advanced or metastatic ER+ disease have encouraged the development of trials in the neoadjuvant setting, although outcomes in this scenario have been mixed (Table 5).

The combination of phosphatidylinositol 3-kinase (PI3K) inhibitors with aromatase inhibitors has not yielded the expected results. The LORELEI trial recruited 334 patients with early-stage ER+ breast cancer, randomized to letrozole + placebo or to letrozole + taselisib. A slight significant difference was found in terms of objective response rates in the taselisib arm (50% vs. 39%, *p* = 0.049), but no significance was observed concerning pCR (1.8% in letrozole + taselisib vs. 0.9% in letrozole + placebo) [98]. Results in the NEO-ORB study, which randomized 257 patients to letrozole + placebo or letrozole + alpelisib, were even more disappointing. Addition of alpelisib to 24-week neoadjuvant letrozole treatment did not improve response in patients with HR+ early breast cancer with respect to pCR, objective response rates, and Ki67 reduction [93].

Good results in the HER-2+ population encouraged the study of the role of Tyrosine Kinase inhibitors in ER+/HER-2 negative. Guarneri et al. conducted a randomized trial that recruited 92 women to receive letrozole plus a placebo or letrozole plus lapatinib [92]. No differences were observed concerning clinical response (63% in the placebo arm and 70% in the lapatinib arm), but significant results were found in the presence of PIK3C mutation (objective response rate, 93% vs. 63% in PIK3C wild-type, *p* = 0.040).

The addition of CDK4/6 inhibitors (palbociclib, ribociclib, and abemaciclib) to NET has been studied in six different trials, with similar results. [94,95,96,99,100,101] The MONALEESA study was a very small (only 14 patients) phase II “window of opportunity” trial that compared Ki67 results after 14 days with letrozole or letrozole + ribociclib (400/600 mg) [101]. A greater Ki67 reduction was observed in the ribociclib arm (69% reduction with placebo vs. 96% and 92 % with ribociclib 400 and 600 mg), as well as a decreased expression of Cyclin D-cyclin-dependent kinase genes. In three trials that combined palbociclib or abemaciclib with NET (NeoPalAna, PALLET, and NeoMONARCH) [94,99,100], a higher complete cell-cycle arrest was observed in the CDK4/6 arm, although little or no benefit was found with regard to pCR. The two remaining trials compared the combination of letrozole with either palbociclib (NeoPAL) [95] or ribociclib (CORALLEEN) [96] with different chemotherapy regimens. The NeoPAL trial randomized 106 patients to either letrozole + palbociclib for 19 weeks or to NCT (5FU 500 mg/m^2^, epirubicin 100 mg/m^2^, cyclophosphamide 500 mg/m^2^ × 3 cycles followed by docetaxel × 3 cycles). Although pCR and Ki67 geometric mean reduction were slightly larger in the CT arm (5.9% vs. 3.8%), the PEPI 0 score was higher in the letrozole-palbociclib subgroup. No differences were observed in terms of BCS rates (69% in both arms). Because of the similarities between the arms, the authors concluded that NET + CDK4/6 inhibitors may represent a safer and less toxic alternative for high-risk ER+ tumors. In the Spanish CORALLEEN trial, 106 patients were randomized to letrozole-ribociclib for six months or to NCT (doxorubicin-cyclophosphamide x 4 followed by weekly docetaxel × 12). The primary endpoint of this study was to evaluate the proportion of patients with PAM50 low risk-of-relapse (ROR) disease at surgery. Most patients belonged to a high-ROR group at baseline (85% and 89%), and at surgery 46.9% (95% CI, 32.5–61.7) of patients in the ribociclib-plus-letrozole group and 46.1% (32.9–61.5) of patients in the chemotherapy group were low ROR; therefore, some patients with high-risk, early-stage ER+ breast cancer could achieve a molecular downstaging with CDK4/6 inhibitors and endocrine therapy.

Finally, the combination of NET with mTOR inhibitors such as everolimus was described by Baselga et al. [97] in a phase II trial in which 270 patients were randomly assigned to receive letrozole + placebo or letrozole + everolimus over four months. Response rate by clinical palpation in the everolimus arm was non-significantly higher than with letrozole alone (68.1% vs. 59.1% *p* = 0.06). Ki67 reduction was also better in the everolimus group (57% vs. 30% *p* < 0.01). No differences in pCR were observed and with evrolimus up to 22.6% of patients presented toxicity (grade 3–4 adverse events).

In conclusion, adding targeted therapies to NET frequently achieves a better cell-cycle arrest and Ki67 drop, although the clinical benefit seems meager. The addition of CDK4/6 inhibitors to NET in high-risk ER+ breast cancer may be comparable to NCT in terms of clinical value and molecular downstaging. In light of the risk-benefit balance, mTOR inhibitors do not appear to be an alternative, due to their toxicity. In any case, further studies are needed, mainly involving long-term survival outcomes.

## 8. The Role of Genomic Assays in NET

Genomic assays have become a very useful tool for guiding adjuvant treatments in early ER+ breast cancer. The possibility of performing these assays in the core biopsy increases their utility to the neoadjuvant setting, and most of them have already demonstrated their capacity for predicting NCT response [102,103,104,105,106,107,108,109]. We will now review those that have been investigated in the context of NET.

### 8.1. MammaPrint^®^/BluePrint^®^

MammaPrint^®^, a 70-gene signature, was the first validated genomic assay and it has been largely used to guide treatments in the adjuvant scenario [110]. BluePrint^®^ is a genomic profile that studies the expression of 80 additional genes, and, combined with MammaPrint^®^, makes it possible to establish a molecular classification of the tumor into the luminal A, luminal B, Her-2, or basal type [111]. The NBRST trial, with over 420 ER+/Her-2 negative patients, demonstrated that the MammaPrint^®^/BluePrint^®^ tandem was more accurate than immunohistochemistry in predicting response to either NCT (with a higher response rate in the basal and Her-2 molecular subtypes) [111] or NET (68% of clinical response to AI in the luminal subtypes) [112]. Hence, luminal breast cancer patients diagnosed by Mammaprint^®^/BluePrint^®^ could be considered for NET, and those with luminal A subtypes have a favorable prognosis. At all events, further prospective research is required.

### 8.2. Oncotype DX

The Oncotype DX^®^ Breast Recurrence Score (RS) assay is a 21-gene validated genomic profile. It was first developed to assess recurrence risk for patients with early ER+/HER-2 negative breast cancer treated with tamoxifen [113] and was initially applied only to node-negative patients, but more recently it has been used regardless of node status [114,115]. The RS stratifies patients into three risk categories: low (<18), intermediate (18–30), and high (>31). In postmenopausal women, a low or intermediate RS profile implies no benefit from additional chemotherapy, while high RS tumors have better outcomes with it [116]. In the neoadjuvant setting, performing Oncotype DX^®^ in the core biopsy has proven to be a useful tool for choosing the best primary therapy so, in those patients with low-intermediate RS, NET might be an effective strategy [108,117]. In a Japanese study by Ueno et al., correlation between RS and NET with exemestane was investigated. Exemestane was given for 24 weeks to 102 patients and RS was analyzed in both the core biopsy and the surgical sample. The clinical response rate in patients with a low or intermediate RS was significantly higher than in those with a high RS (59.4% and 58.8% vs. 20.0% *p* = 0.015). Differences were also observed in BCS rates, which were higher in the low (90.6%) and intermediate (76.5%) RS cases and lower in the high RS group (46.8%, *p* = 0.028) [118]. The phase III TransNEOS trial evaluated data from 295 postmenopausal ER+, node-negative breast cancer patients treated primarily with letrozole for six months [117]. The primary endpoint was to establish differences in clinical response rates according to RS (low vs. high). A larger proportion of patients with low RS had clinical responses (54%), compared to those with high RS (22%, *p* < 0.001). In multivariable analyses, a continuous RS result was associated with clinical response (*p* < 0.001). An RS <18 was also associated with better BCS rates (*p* = 0.010). These results confirmed the capacity of this genomic profile to predict the clinical response to neoadjuvant letrozole. Therefore, Oncotype DX^®^ RS has proved itself to be a very helpful tool for determining which neoadjuvant treatment is better with regard to RS, the lower scores being more suitable for NET and the higher ones (>31) better candidates for NCT.

### 8.3. EndoPredict^®^

EndoPredict^®^ (EP) is a 12-gene genomic assay that provides prognostic information on the risk of a distant recurrence in breast cancer patients [119], and it is valid for both negative- and positive-node status. EP prognostic power has been improved by creating EPclin: a risk score that incorporates pathological data like tumor size or nodal status. However, in the neoadjuvant setting, only the molecular analysis (EP) is possible. Dubsky et al. analyzed 217 samples of diagnostic core biopsies from the ABCSG-34 trial [106], in which women had already been selected to receive NCT or NET according to clinical or immunohistochemical criteria. Tumors with a low molecular score (MS) were unlikely to benefit from NCT (NPV 100% (95% CI, 66.4%–100%)), whereas a high MS predicted resistance to NET (NPV 92.3% (95% CI, 79.1%–98.4%)). EP MS predicted a residual cancer burden after treatment with neoadjuvant therapies for patients with HR-positive, HER2-negative early-stage breast cancer. On the other hand, this genomic assay may also have a part to play in estimating prognosis after treatment with CDK4/6 inhibitors. In the N007 study, the aim of the authors was to evaluate PEPI score, pathologic response, and gene expression on the tumor samples of 20 patients who were treated for 24 weeks with letrozole + Palbociclib [120]. EPclin was found to be a better parameter than the PEPI score for estimating a prognosis, since some of the high PEPI-score risks had a low-risk EPclin. Furthermore, changes in gene expression after letrozole + palbociclib were reported, implying that not only proliferative activity was reduced but also immune signaling was modified. The biological information provided by EndoPredict^®^ can therefore facilitate patient selection for neoadjuvant treatments, and EPclin might be a more accurate biomarker than PEPI score, especially in NET treatments combined with CDK4/6 inhibitors. In any case, additional studies will be necessary in order to confirm these results.

### 8.4. A Four-Gene Model

Turnbull et al. developed a four-gene model from pretreatment and on-treatment samples of 89 postmenopausal women who were receiving NET with letrozole [121].

With an accuracy of 96% for predicting response status, this model was created on the basis of the level of two genes before treatment (IL6ST, associated with immune signaling, and NGFRAP1, related to apoptosis) and on the levels of two proliferation genes (ASPM, MCM4) after two weeks of treatment. A further independent validation was performed in a similar cohort treated with anastrozole, with 91% accuracy. The predictive power was increased when the two-week on-treatment biopsies were added to the pretreatment biopsies. The four-gene signature also significantly predicted recurrence-free survival (*p* = 0.029) and breast cancer–specific survival (*p* = 0.009), and it is feasible that the results can be obtained by means of PCR and immunohistochemistry techniques. This approach offers the advantage of taking into account the dynamic changes produced by NET, suggesting that a full standard course of several months of neoadjuvant treatment may not be required, and a short two-week pre-surgical therapy might be sufficient to improve the accuracy of long-term predictions of outcomes [122]. Nevertheless, these results are based on retrospective analysis, so larger prospective studies should be conducted.

### 8.5. Other Gene Expression Profiles

Several research groups have described different forms of gene expression related to NET response. The Italian group led by Mello-Grand et al. defined an expression signature of 54 genes that predicted response in 17 patients treated with anastrozole for three months [123]. In responding tumors, down-regulated genes after treatment were mainly related to cell cycle, many of them transcriptionally activated by ESR1 (estrogen receptor1), whereas up-regulated genes were associated with immune response, histamine metabolism and inflammation, cell adhesion, and cytoskeleton. Nonresponding tumors showed an increased immune response as well as androgen receptor nuclear signaling, and this might explain endocrine resistance. Five of these genes were found to predict response on an independent dataset of 52 breast cancers treated with letrozole (*p* = 0.0056). Miller et al. developed a 99-gene signature that evaluated long-term estrogen-deprived (LTED) gene expression [124]. The genes up- or down-regulated in LTED cells were expressed at higher or lower levels, respectively, in tumors where cell proliferation was not suppressed by letrozole. These results were later validated in another dataset of patients treated with anastrozole. The LTED gene expression signature was found to be predictive of high tumor-cell proliferation following neoadjuvant therapy with anastrozole and letrozole, therefore making it possible to identify prospectively patients at risk of early recurrence during or following ET. Xu-Liang et al. performed gene expression profiling on 86 pre-NET and post-NET tumor samples from the CARMINA trial, which randomized patients to either letrozole or fulvestrant [125]. In NET responders, involved immune-associated genes enriched in activated Th1 pathway were found, but these remained unchanged in nonresponders. Gene expression also revealed that lipid metabolism was the main molecular function related to prognosis. Finally, the cell cycle-apoptosis pathway and the PIK3CA/AKT/mTOR pathway were altered significantly more frequently in nonresponders than in responders. Inda et al. have recently described an ER pathway activity score (ERPAS) involving the expression of 27 high-evidence target genes of the ER transcription factor in pre- and post-treatment tumors treated with letrozole, exemestane, or fulvestrant [126]. One third of the ER+ breast cancers presented an inactive ERPAS at baseline. Highly respondent tumors had a higher pretreatment ERPAS and a greater magnitude of decrease after two weeks of therapy. As a result, this score may serve to identify ER+ tumors that will not respond to ET.

To sum up, genomic profiling performed on diagnostic core biopsies as well as on early on-treatment biopsies may become a very useful tool for enabling NET response or endocrine resistance to be predicted. Larger prospective studies will be necessary in order to establish which group of genes can provide the maximum information, and therefore act as a reliable biomarker.

## 9. Ongoing Trials

There are currently numerous clinical trials focusing on NET (Table 6), the majority of which involve the combination of NET with targeted therapies.

At least seven of these uses any of the CDK4/6 inhibitors together with an AI or tamoxifen. Other biological molecules such as PI3K inhibitors, tyrosine kinase inhibitors [127], anti-VEGF, Ros inhibitors, or histone deacetylase inhibitors, combined with an AI, are also being studied. It is worth highlighting the great interest of the immune role in ER+ breast cancer, especially the antitumor immune response mediated by the programmed death 1/programmed death ligand 1 (PD1/PD-L1) and cytotoxic T lymphocyte antigen-4 inhibitors. Anti-PD-1/PD-L1 are already emerging as a promising treatment in non-luminal phenotypes and there are, to our knowledge, three clinical trials (NCT03573648, NCT02997995, NCT03874325) that are studying avelumab or durvalumab, combined with an AI, in the ER+ breast cancer neoadjuvant setting.

Genomic assays are being used for screening patients for various neoadjuvant alternatives. MammaPrint is the basis of the NCT03900637 trial, which randomizes ER+ breast cancer patients who are not suitable for BCS to either NCT or NET. Oncotype DX is being used in the DxCARTES trial to select candidates for NET with letrozole or letrozole + palbociclib.

Invasive lobular carcinoma accounts for only 15% of breast cancers, but it is known that these are usually especially responsive to NET, even though studies involving this histological subtype are scarce [128]. Only one “window of opportunity” trial focuses on the characteristics of NET in invasive lobular tumors (NCT02206984), comparing Ki67 changes and ER/PR expression after a 21-day cycle of tamoxifen or anastrozole or two doses of fulvestrant.

Finally, premenopausal patients continue to be a subject of concern, since as yet there is insufficient evidence of NET’s safety for this group. Several of the studies mentioned previously recruited premenopausal women, and they are also the exclusive focus of two Chinese trials (NCT0339753, NCT02535221) that compare NCT with NET in younger patients.

## 10. Conclusions

The evidence available to date shows that NET is an effective and safe alternative for the treatment of ER+/Her-2 negative postmenopausal patients, enabling higher BCS rates to be achieved, together with survival rates equivalent to those provided by NCT, even in cases of low axillary burden. NET has also proven to offer unquestionable opportunities for testing new therapeutic agents. In the future, the role of CDK4/6 inhibitors, as well as other target treatments, will probably extend NET indications to high-risk ER+ breast cancer patients. Nevertheless, a few questions remain unclear. In the first place, treatment duration, although usually recommended for 4–6 months, might sometimes be insufficient, and the optimal treatment length has yet to be determined, and probably needs to be individualized. Second, the innate endocrine resistance of some ER+ tumors raises concerns about the proper selection of patients. In this regard, genomic assays and other multigene tests performed on the diagnostic biopsy provide a very useful tool for determining endocrine sensitivity. Early on-treatment biopsies and Ki67 analysis may also offer valuable information about treatment response as well as prognosis; however, more reliable biomarkers should be investigated. Finally, offering NET to premenopausal women still remains controversial because there has been too little research on this group. Nevertheless, large luminal A-like tumors may be safely managed in this way in order to achieve BCS. The results of trials now in progress, as well as further research, will help broaden NET indications for these women.

## Figures and Tables

**Table 1 cancers-13-02538-t001:** Clinical trials comparing NET vs. NCT.

Clinical Trial	No	Characteristics	Chemotherapy	Endocrine Therapy	Primary Endpoint	Response	BCS Rate
Semiglazov et al., 2007 [15]	239	Postmenopausal ER+ and/or PR + stage IIA − IIIB	Doxorubicin+ paclitaxel × 4 cycles	Anastrozole/ Exemestane × 12 wk	CR	64% CT vs. 64% ET	24% CT vs. 33% ET
Alba et al., 2012 (GEICAM 2006-03) [13]	95	Pre and postmenopausal ER+/PR+/HER2−	EC × 4, followed by docetaxel × 4	Exemestane × 24 wk (+goserelin if premenopausal)	OR (RECIST, MRI)	66% CT vs. 48% ET	47% CT vs. 56% ET
Palmieri et al., 2014 (NEOCENT) [16]	44	Postmenopausal ER+	5-FU + EC × 6, switched to docetaxel after 3 cycles if stability/ progression	Letrozole × 18 wk	OR (US, mammogram)	55% CT vs. 59% ET	55% CT vs. 68% ET

ER = Estrogen receptor; PR = Progesteron receptor; CR = clinical response; CT = chemotherapy; EC = epirubicin + cyclophosphamide; ET = endocrine therapy; OR = objective response; US = ultrasound, wk = week.

**Table 2 cancers-13-02538-t002:** Studies comparing tamoxifen vs. aromatase inhibitors vs. fulvestrant.

Clinical Trial	No of Patients	Duration (months)	Drugs	Outcomes
P024 (2001) [7]	337	4	Letrozole vs. TMX	CRR 55% vs. 36% (*p* < 0.001) RRR 35% vs. 25% (*p* = 0.042) BCS 45% vs. 35% (*p* = 0.022)
PROACT (2005) [46]	451	3	Anastrozole vs. TMX	CRR 50.0% vs. 46.2% (*p* = 0.37) RRR 39.5% vs. 35.4% (*p* = 0.29) BCS: 43% vs. 30.8% (*p* = 0.04)
IMPACT (2006) [8]	330	3	Anastrozole vs. TMX vs. Anastrozole + TMX	CRR 38% vs. 36% vs. 39% (ns) RRR 24% vs. 20% vs. 28% BCS: 43% vs. 30.8% vs. 24% (NS)
SEMIGLAZOV [47]	151	3	Exemestane vs. TMX	OR 76% vs. 40% (*p* = 0.05) BCS 37% vs. 20% (*p* = 0.05)
STAGE (2012) [12]	197	6	Anastrozole + Goserelin vs. TMX + Goserelin	CRR 70.4% vs. 50.5% (*p* = 0.004) RRR 58.2% vs. 42.4%, (*p* = 0.027) BCS 86% vs. 68%
ACOSOG Z1031 (2011) [48]	377	4	Exemestane vs. Letrozole vs. Anastrozole	CRR 62.9% vs. 74.8% vs. 69.1% (NS)
CARMINA (2016) [49]	116	4 or 6	Anastrozole vs. Fulvestrant	CRR 52.6% vs. 36.8% (NS) BCS 57.6% vs. 50% (NS)
QUENEL-TUEUX (2016) [50]	108	6	Anastrozole vs. Fulvestrant	CRR 59% vs. 54% (NS) BCS 59% vs. 50%

TMX = tamoxifen; CRR = clinical response rate; OR = objective response; RRR = radiological response rate; BCS = breast conserving surgery, ns = non-significant.

**Table 3 cancers-13-02538-t003:** Duration of NET.

Study	No of Patients	Type of NET	Duration (months)	Assessment	Outcomes
Krainick-Strobel et al. 2008 [50]	32	Letrozole	4 to 8	Monthly: palpation 3 monthly: MRI, Mammogram, or US	4 mo 55% ORR 71% BCS 8 mo 72.4% ORR/>80% BCS
Dixon et al. 2009 [49]	182	Letrozole	3 vs. >3	Clinical and US measurement (0,2,6,12 wk) Mammogram (0.12 wk) Review/3 mo	3 mo: 70% ORR 60% BCS >3 mo: 83% ORR 72% BCS
Llombart-Cussac et al. 2012 [51]	70	Letrozole	4–12	Monthly: clinical examination Mammogram and US/8wk for first 4 mo	76.8% ORR (25% CR and 51.8% PR) 43% BCS
Allevi et al. 2013 [54]	120	Letrozole	4,8 or 12	Monthly: clinical palpation (caliper) Mammogram and US at baseline and before surgery	4 mo: pCR 2.5%; ORR 45%; BCS 80% 8 mo: pCR 5%; ORR 86.8%; BCS 85% 12 mo: pCR 17.5%; ORR 95%; BCS 87.5%
Hojo et al. 2013 [56]	52	Exemestane	4 vs. 6	Monthly: caliper measurement and toxicity assessment Ultrasound and Mammogram if progression suspected	4 mo: pCR 0%; ORR 42.3%; BCS 50% 6 mo: pCR 4 %; ORR 48%; BCS 48%
Carpenter et al. 2014 [52]	139	Letrozole	up to 12	Clinical examination and bimodal US/2 mo until BCS	ORR 85% (3.2% CR and 81.5% PR) 66% BCS Median time to achieve tumor response to allow BCS: 7.5 mo
Fontein et al. 2014 [53]	102	Exemestane	3 to 6	Monthly: clinical palpation 3 monthly: MRI, Mammogram, or US	pCR: 0.98% 3 mo: ORR 58.7% BCS 58.7% >3 mo: ORR 68.3%; BCS 70.6%
Rusz et al. 2015 [55]	42	Letrozole	12	Clinical palpation every 3 months. Imaging as necessary	pCR: 14.3% operated cases ORR: 88% BCS: 45%

Mo = month; wk = week; US = ultrasound; pCR = pathological Complete Response; ORR = Objective Response Rate; CR = Clinical Response; PR = Partial Response; BCS = breast-conserving surgery.

**Table 4 cancers-13-02538-t004:** PEPI score.

Pathological Characteristics of Surgical Specimen	RFS		BCSS	
	HR	Score	HR	Score
Tumor size T1/2 T3/4	- 2.8	0 3	- 4.4	0 3
Nodal status Negative Positive	- 3.2	0 3	- 3.9	0 3
Ki67 level 0%–2.7% >2.7%–7.3% >7.3%–19.7% >19.7%–53.1% >53.1%	- 1.3 1.7 2.2 2.9	0 1 1 2 3	- 1.4 2.0 2.7 3.8	0 1 2 3 3
ER status (Allred score) 0–2 3–8	2.8 -	3 0	7.0 -	3 0

With permission from Ellis M. et al. [85]. HR = hazard ratio; RFS = relapse free survival; BCSS = breast cancer specific survival.

**Table 5 cancers-13-02538-t005:** NET and Targeted Therapies. Phase II trials.

Drug Group	Study	No of Patients	AI	Targeted Therapy	Design	Outcomes	pCR
PI3K Inhibitors	LORELEI [88]	334	L	Taselisib (T)	L + T/L + P	ORR 50%/39% *p* = 0.049	1.8%/0.6% ns
	NEO-ORB [89]	257	L	Alpelisib (Al)	L + P/L + Al	ORR 61.0%/63.4% ns	1.7%/2.8% ns
Tyr Kin inhibitors	Guarneri et al. [91]	92	L	Lapatinib (Lp)	L + P/L + Lp	ORR 63%/70% ns	93% ORR in PI3K mut
CDK4/6 inhibitors	MONALEESA-1 [90]	14	L	Ribociclib 400 mg (R400) 600 mg (R600)	L 2 wk/ L + R400 2 wk/ L + R600 2 wk	Higher Ki67 reduction in the R arm	-
	NeoPalAna [92]	50	A	Palbociclib (Pal)	A → A + Pal	CCCA higher after adding Pal C1D15/C1D1 87%/26%, *p* < 0.001	-
	NeoPAL [93]	106	L	Palbociclib (Pal)	L + Pal 19 wk vs. FEC/21 d × 3 + docetaxel/21 d × 3	RCB higher in the CT arm BCS equal in both arms PEPI 0 17.6%/8.0%	3.8%/5.9%
	NeoMONARCH [94]	224	A	Abemaciclib (Ab)	A/Ab/Ab + A 2 wk and 2nd core biopsy Ab + A 14 wk	CCCA higher in Ab arms 14%/58%/68% *p* < 0.001	4%
	PALLET [95]	307	L	Palbociclib (Pal)	L 14 wk/ L 2 wk → L + Pal 14 wk Pal 2 wk → L + Pal 14 wk L + Pal 14 wk	Ki67 reduction and CCCA higher in the L + Pal arm 90%/59%. *p* < 0.001 CR 49.5%/54.3% ns	ns 1.1% in L vs. 3.3% in L + Pal
	CORALLEEN [96]	106	L	Ribociclib (R)	L + R 28 d × 6/ DC/21 d × 4 → Paclitaxel/7 d × 12	ROR at surgery 46.9%/46.1%	2%/3%
mTOR inhibitor	Baselga et al. [97]	270	L	Everolimus (E)	L/L + E 4 mo	CR 59.1%/68.1% ns	0.8/1.4% ns

AI = Aromatase Inhibitor; L = Letrozole; A = Anastrozole; ORR = objective response rate; ns = non-significant; wk = week; mo = month; CCCA = complete cell cycle arrest (central Ki67 <2.7%); C1D1 = cycle 1 day 1; C1D15 = cycle 1 day 15; FEC = 5FU 500 mg/m^2^ + epirubicin 100 mg/m^2^ + cyclophosphamide 500 mg/m^2^; RCB = Residual Cancer Burden; CT = chemotherapy; CR = Clinical response; DC = doxorubicin + cyclophosphamide; CR = clinical response; → = followed by.

**Table 6 cancers-13-02538-t006:** Ongoing Trials with targeted therapies.

Targeted Therapy	Study	Population	Design	Arms	Primary Endpoints	Secondary Endpoints	Status
CDK4/6inhibitor	NCT04293393(CARABELA)	Stages II-IIISurgery feasible	PhaseII	Arm A:AC/21 days × 4 cycles → Paclitaxel/7 d × 12 wk or3-weekly docetaxel/21 d × 4 cycles (wk)Arm B:letrozole plus abemaciclib ± LHRH up to 12 mo	RCB	Ki67 changesRCB 0+ I versus RCB-II versus RCB-IIIPEPI scoreIEFSMolecular downstaging for high-risk genomic groups	Recruiting
NCT03969121	T > 15 mmKi67 > 14%	PhaseIII	Arm A:Letrozole (+LHRH if premenopausal) + placebo 16 wkArm B:Letrozole(+LHRH if premenopausal) + palbociclib 16 wk	PEPI scoreEPclin score	CRRKi67 changepCRBCSAdverse events	Recruiting
NCT03819010(DxCARTES)	T > 2 cmKi67 ≥ 20%	PhaseII	Pretreatment RS 18.25:letrozole (+LHRH if premenopausal) + palbociclib/6 cyclesPretreatment RS 26-100:letrozole (+LHRH if premenopausal) + palbociclib/6 cycles	Difference on RS pre- and post-treatment (molecular results)	Molecular changesconcordance rate between the RCB score (0–I vs. II–III) and pCR and post-treatment RS	Completed
NCT03065621(NeoRHEA)	T2–T3 N0–N1	PhaseII	Single arm:palbociclib 125 mg × 4 cycles + letrozole/tamoxifen	Biomarkersof resistance	Radiological response	Completed
NCT02712723	ER+ >66%/Allred score 6–8Stage II-III	PhaseII	Arm A: letrozole + placebo 22 wkArm B: letrozole + ribociclib 600 mg 22 wkArm C: letrozole + ribocilib 400 mg 22 wk	PEPI score 0	CCCApCRBCSRFS	Active, not recruiting
NCT02603679(PREDIXLumB)	Luminal B any NLuminal A N+	Phase II	Arm A: Weekly paclitaxel 12 wk → switch to NET + palbociclibArm B: tamoxifen + palbociclib 12 wkArm C: AI + palbociclib 12 wk switch toArm D: goserelin + AI + palbociclib 12 wk weekly paclitaxel	RRR after 12 wk	pORRBCSRFSBCSSOS	Recruiting
NCT02592083(PREDIXLumA)	Luminal AIDC>40 y	PhaseII	Arm A:ET (tamoxifen or AI or AI + goserelin)Arm B:ET + palbociclib	CRRRRR	pORRBCSRFSBCSSOS	Recruiting
*Immunotherapy*	NCT03573648	Stage II-III	PhaseII	Arm A:ETArm B:ET + palbociclib (PET) in a 1:2 ratio.After 1 cycle (1 mo) both arms will receive avelumab (A) × 3 cycles	CCR	Adverse events	Recruiting
NCT02997995	PostmenopausalLuminal A	PhaseII	1st phase:immune attractant + exemestane 6 wk.After three wk (± 3 days), a tumor biopsy2nd phase:Patients > 10% CD8+ cells in the tumor after 3 wka durvalumab 1500 mg Q4W IV,combined with exemestane (25 mg daily), for six months.	pCR	Number of CD8+ T cell Clinical responseAssessment of Ki67ToxicitiesPredictive value of Mutational load for efficacy of DurvalumabPredictive value of PDL1	Completed
NCT03874325	T2–T4c,any N	PhaseII	Single arm:1500 mg durvalumab i.v/4 wk × 6 cycles + anastrozole 1 mg	PEPI score = 0	CRRCPDCSD	Active, no recruiting
*Anti-VEGF*	NCT00773695	Her-2neg BCT > 25 mm	PhaseII	CT arm:FEC 12 wk → taxane × 12 wkFEC 12 wk → taxane × 12 wk + bevacizumabET arm: AI 24 wkAI + bevacizumab	pCR	pORRtype of surgery	Active,recruiting
*PI3K inh*	NCT01275859	PostmenopausalT2N0-3	PhaseII	Single arm:letrozole 2.5 mg + lapatinib 1500 mg po for 18–21 wk	pCR	CRRRRRDFSOS	Completed
*Tyr Kin inh*	NCT02562118	ER+Postmenopausal	PhaseI–II	Single arm:lenvatinib × 2 wk → letrozole 2.5 mg daily + lenvatinib × 12 wk	CRR	pCRPFS	Recruiting
*Ros inh.*	NCT04551495(ROSALINE)	LIC neg	PhaseII	Single arm:428-day of letrozole 2.5 mg + entrectinib 600 mg daily.(+goserelin if premenopausal)	RCB	pCRORR (by MRI)Adverse events	Not yetrecruiting
*Histone* *deacetylase inh.*	NCT04465097	Stage II/III	PhaseII	Single arm:exemestane from week 1 to week 26and tucidinostat BIW from week 3 to week 26(+leuprorelin/goserelin if premenopausal)	ORRevaluated by MRI	ORR evaluated by USpCRAdverse eventsRCB	Recruiting

**Other Ongoing Trials**
Lobular breast cancer	NCT02206984	T > 1 cm	PhaseII(WoT)	Arm A: tamoxifen 21 dArm B: anastrozole 21 dArm C: fulvestrant day 1 and 14	Change in Ki67	ER expressionER gene expressionPR expression	Recruiting
	NCT03397537	T > 1 cm	PhaseII	letrozole 2.5 mg × 6 mo(+aGnRH if premenopausal)	ORR	-	Completed
Premenopausal	NCT02535221	>35 < 55 yT2N0M0	PhaseIII	Arm A: goserelin + TAM + AI (1st 4 wk TAM → AI)Arm B: FEC × 4–6 cycles	RRR by US	pCR (Miller-Paine)	Recruiting
*MammaPrint* ^®^	NCT03900637	Stage I–IIIABCS not feasible	PhaseII	MammaPrint^®^ high risk: NCT × 4 followed by docetaxel × 4MammaPrint^®^ low risk: letrozole 2.5 mg × 16–24 wk(+leuprorelin if premenopausal)	BCS conversion rate	pCRCRRTumor size reduction rateDFSIBTRBluePrint^®^	Recruiting

BC = breast cancer; ER = estrogen receptor; HR = hormone receptor; AC = Adriamycin-Cyclophosphamide; Inh = inhibitor; wk = week; mo = month; IDC = infiltrating ductal carcinoma; ILC = invasive lobular carcinoma; BCS= breast conserving surgery; IEFS = invasive event free-survival; CCCA = complete cell cycle arrest; RFS = relapse free survival; PFS: progression free survival; NCT = neoadjuvant chemotherapy; ET = endocrine therapy; AC = Adriamycin/Cyclophosphamide; FEC = Fluorouracil + epirubicin + cyclophosphamide; CCR = clinical complete response; CRR = clinical response rate; RRR = radiological response rate; DFS = disease free survival; IBTR = ipsilateral breast tumor recurrence; CPD = clinical progressive disease; CSD = clinical stable disease; ORR = overall response rate; pORR = pathological objective response rate; pCR = pathological complete response; RCB = residual cancer burden; WoT = window of opportunity trial; IHC = immunohistochemistry; TAM = tamoxifen; AI = aromatase inhibitor; aGnRH = gonadotropin-releasing hormone agonist; → = followed by.

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
