# Peer review of "The Present and Future of Neoadjuvant Endocrine Therapy for Breast Cancer Treatment"

_cancers, 2021, doi:10.3390/cancers13112538_

Round 1

Reviewer 1 Report

The authors have nicely summarized current evidence with a critical view. They describe the criteria for patient selection, therapy combination, therapy duration and monitoring of the response, as well as the usefulness of gene expression signatures. The paper is well structured and written, this makes a useful source of information for clinicians and researchers in the breast cancer  field.

Author Response

Thank you very much for your comments. Small corrections have been made to the text.

Kind regards

Reviewer 2 Report

This manuscript is a well-organized and well-written review of the current status and future prospects of neoadjuvant endocrine therapy (NET) for primary breast cancer. Data from previous clinical trials and information from ongoing trials are sufficient and should be of interest to all oncologists treating breast cancer patients. A few concerns need to be addressed.

Given the difference in endpoints between neoadjuvant chemotherapy (NCT) and NET, it is not necessary to aim for pCR because pCR is not a surrogate marker for clinical outcome with NET. Rather, response or sensitivity to NET is an important determinant of long-term clinical outcome in patients with luminal breast cancer. Therefore, there are several ways to determine preoperative endocrine sensitivity using multigene gene assays and Ki-67 as windows of opportunity after NET, but the most important thing seems to be how to overcome endocrine resistance leading to reduced distant metastasis during the follow-up period for complete cure.

In fact, NET is less toxic than NCT and is useful in increasing BCS, but there is no difference in survival of postmenopausal patients. Given that there is no difference in overall survival (OS) between patients who received ET preoperatively and those who received ET postoperatively, an important advantage of NETs is the detection of endocrine resistance and strategies to overcome resistance and alternative treatments. These strategies can lead to cure of primary breast cancer of luminal type. Mechanism of endocrine resistance and strategies to overcome them are essential for cure.

In view of overcoming resistance to ET, it is of interest what functional role immune cells such as tumor infiltrating lymphocytes (TILs) play in host defense immunity and long-term immunity for cure. Particularly, TILs, which are abundantly present in luminal tumor cells, may contain regulatory T cells for immunosuppression. Genetic mutations and persistent dormant cells regulated by antitumor immunity during endocrine therapy may be important for therapeutic effect and prognosis.

Author Response

Thank you very much for your comments

  1. We agree, and it is reflected in the manuscript, that pCR should not be and endpoint with NET. We describe other biomarkers that are more related to NET efficacy as well as prognosis.
  2. NET offers several advantages. It does not improve survival, but it improves BCS rates. In addition, we agree with you that it may be a very useful tool to detect resistance, either innate or acquired. We mention that mainly when speaking about changes in Ki67. 
  3. It is very interesting the role of TILs in breast cancer although, to our knowledge, it is not clear how they act in luminal BC (we have read papers finding increased TILs described as an adverse prognostic factor- Denkert et al. 2018, and some others as a favorable one-Kolberg-Liedtke et al. 2020). Nevertheless, we haven't found information concerning the role of TILs in NET, although it may be an interesting subject of study.

Thank you again for your appreciations. Kind regards

Reviewer 3 Report

Title: The Present and Future of Neoadjuvant Endocrine Therapy for Breast Cancer Treatment. Manuscript ID: cancers-1196584

It is an interesting review of importance in its field; the study is original.

This review provides an update on the current state of art regarding the present and the future role of  neoadjuvant endocrine treatment.

Abstract is appropriate.

Introduction and background informations are complete; overall presentation is adequate and linear.

Conclusions are linear and appropriate. 

English language and style are of good quality; they are fine/minor spell check required.

Accuracy of literature citations is good. The Authors could update the references with some articles published on 2021, as:

Lerebours F, Cabel L, Pierga JY. Neoadjuvant Endocrine Therapy in Breast Cancer Management: State of the Art. Cancers (Basel). 2021 Feb 21;13(4):902. doi: 10.3390/cancers13040902.

da Silva LR, de Andrade CA, Brenelli F, Ramalho S, Reinert T, de Souza ABA, da Silva AER, de Paula Leite Kraft MB, de Vasconcelos VCA, Frasson AL, Torresan RZ, Cabello C, Ellis MJ, Zeferino LC. Real-world data on neoadjuvant endocrine therapy in ER-positive/HER2-negative breast cancer. Breast Cancer Res Treat. 2021 Apr;186(3):753-760. doi: 10.1007/s10549-020-06076-5. 

The manuscript may be considered eligible to publication with minor revisions.

Author Response

Thank you very much for your remarks and the references suggested. None of these articles were published when this manuscript was written. Nevertheless, we will include the Brazilian paper (the French one is quite similar to ours...)

Small corrections in the text have also been done.

Thank you again for your appreciations. Kind regards